# Biomaterials and Meniscal Lesions: Current Concepts and Future Perspective

**DOI:** 10.3390/pharmaceutics13111886

**Published:** 2021-11-07

**Authors:** Michele D. M. Lombardo, Laura Mangiavini, Giuseppe M. Peretti

**Affiliations:** 1Residency Program in Orthopedics and Traumatology, University of Milan, 20122 Milan, Italy; mdm.lombardo@gmail.com; 2IRCCS Istituto Ortopedico Galeazzi, 20100 Milan, Italy; laura.mangiavini@unimi.it; 3Department of Biomedical Science for Health, University of Milan, 20133 Milan, Italy

**Keywords:** meniscus, biomaterial, scaffold, polymer, bioprinting, stem cells, bioengineering

## Abstract

Menisci are crucial structures for knee homeostasis. After a meniscal lesion, the golden rule, now, is to save as much meniscus as possible; only the meniscus tissue that is identified as unrepairable should be excised, and meniscal sutures find more and more indications. Several different methods have been proposed to improve meniscal healing. They include very basic techniques, such as needling, abrasion, trephination and gluing, or more complex methods, such as synovial flaps, meniscal wrapping or the application of fibrin clots. Basic research of meniscal substitutes has also become very active in the last decades. The aim of this literature review is to analyze possible therapeutic and surgical options that go beyond traditional meniscal surgery: from scaffolds, which are made of different kind of polymers, such as natural, synthetic or hydrogel components, to new technologies, such as 3-D printing construct or hybrid biomaterials made of scaffolds and specific cells. These recent advances show that there is great interest in the development of new materials for meniscal reconstruction and that, with the development of new biomaterials, there will be the possibility of better management of meniscal injuries

## 1. Introduction

Menisci are fibrocartilaginous structures embedded within the knee joint. They were defined in 1897 as “functionless remnants of leg muscle origin” [1]; today, they are considered as structures of fundamental importance for maintaining the integrity of the knee joint homeostasis. Among their functions, we must remember the increase in congruence between the articular surfaces of the distal femur and tibial plateau, bear loading, shock absorption, lubrication and proprioception [2]. Due to their anatomical position, they are susceptible to frequent injuries, especially in activities where changes in direction are expected, or in the case of contact sports [3]. Given the poor vascularization of the menisci, especially in the innermost portions of the joint, the injuries are characterized by a low healing potential. In the early days of meniscal surgery, a complete removal of the meniscus was the gold standard treatment. This procedure, although resolving the symptoms in the short term, led in a few years to the development of osteoarthritis (OA) of the knee [4].

Nowadays, selective arthroscopic meniscectomy remains a treatment of choice if the lesion cannot be subjected to a meniscal suture or repair, which is currently considered the standard treatment for meniscal injuries. 

Regenerative medicine has invested much effort into the treatment of meniscal tears by implementing strategies to restore the highly specialized functions of these structures.

In this review, we will analyze all of the possible treatment strategies that go beyond traditional meniscal surgery: particular attention will be paid to the analysis of the different biomaterials and their bioengineering features, which are under investigation at a basic research level, or are utilized in preclinical research and clinical practice.

We will begin the review by describing the anatomy, function and consequences of a meniscal injury. Then, we will delve into the therapeutic possibilities, giving an overview of the surgical strategies currently in use, and then focusing on possible reconstructive strategies based on the use of meniscal scaffolds. For the discussion, biomaterials will be divided according to their origin, i.e., whether synthetic or biological. Hydrogel, decellularized scaffolds and hybrid scaffolds will be considered. Finally, we will discuss new technologies currently in the field for scaffold generation, such as the 3D printer and the combination of scaffolds with mesenchymal stem cells.

## 2. Current Concepts on Meniscus Anatomy, Structure, Lesions and Their Treatment

### 2.1. Meniscal Anatomy and Function

The menisci are fibrocartilaginous structures present inside the knee, located between the femoral condyles and the tibial plateau. They are divided into medial and lateral. They possess a crescent shape and a triangular cross-section [5] (Figure 1).

Their function is fundamental for knee homeostasis; in fact, with their proximal concavity, they accommodate the femoral condyles with increasing the congruence between the articular surfaces, which increases the stability of the joint; they also have the function of shock absorber, lubrication, proprioception and distribution of loads during movement. Overall, the menisci cover approximately two thirds of the joint capsule. The menisci are held in place by a series of ligaments and attachments to the joint capsule, from which, vascularization derives [6].

Classically, the menisci are divided into three portions based on their vascularization: starting from the most peripheral portion, we find the red-red zone, the red-white zone and the white-white zone. This classification has an important prognostic value; in fact, it reflects the healing potential of a lesion: the greater the vascularity, the greater the possibility of healing. The medial meniscus is vascularized in peripheral 10–30% and the lateral meniscus is vascularized in 10–25% of its width. The more peripheral regions are more prone to healing when they are stabilized with anchors or sutures. We must also remember the existence of a microvascular plexus originating at the level of the joint capsule and the synovial membrane surrounding the menisci. Similarly, the poor vascularization does not allow for the recall of healing cells in the event of injury: this is particularly true in the case of lesions at the level of the innermost portions of the meniscus, which are completely avascular [7] (Figure 2).

The menisci also have the important function of augmenting the congruence between the femoral and tibial condyles, increasing the joint compliance, which is a requirement for the lubrication of the fluid film and normal circulation of synovial fluid within the knee [8].

Unlike articular cartilage, the menisci have intrinsic innervation, which is important for the proprioception of the knee. The innervation is richly represented at the level of the horns, both anterior and posterior: this reflects the fact that, during the movement arc, information is sent centrally, which helps to define the position of the knee in space and to trigger an arch reflex that stimulates postural muscle tone in a protective sense [7].

### 2.2. Meniscal Structure

A total of 75% of the meniscus weight is composed of water. Regarding the dry weight, approximately 60–70% is made up of collagen, 1% is made up of proteoglycan and 8-13% is made up of non-collagen proteins, such as elastin [9]. Collagen fibers, of which, type I is clearly the most frequent (approximately 90%), are predominantly arranged in a circumferential way, following the C-shape of the meniscus, while a minor part is radially oriented [10]. The innermost zone is characterized by chondrocyte-like cells, and there is a higher prevalence of type II collagen. The outermost area, on the other hand, has many fibroblast-like elongated cells. Furthermore, in the most superficial portion, there are cells responsible for producing proteins with a lubricating and anti-adhesive function, and progenitor cells that have a regenerative potential [11].

The collagen fibers within the menisci are arranged into three distinct layers. Most of the fibers are located in the middle layer and are circumferentially oriented, offering resistance to hoop stresses [12]. This layer is sandwiched between two surface layers, in which, there are shorter radially arranged fibers that act as bonds, providing structural rigidity against compressive forces and preventing longitudinal cleavage, as well as resistance to shear forces [13,14] (Figure 3).

The biomechanical properties of the meniscus are given, point by point, by the molecular composition and by the microstructure of the tissue: approximately 50% of the load of the knee passes through the menisci when it is in extension, which reaches 85% when the knee is in the flexion position. Thus, under a load, in addition to tensile and shear forces, the meniscal structures must also oppose radial forces that tend to extrude the meniscus [15].

It is therefore clear that the meniscus is essential for knee homeostasis and protection against the development of knee arthritis, which is why many efforts must be directed towards the development of new technologies.

### 2.3. Meniscal Lesions and Treatment

Meniscal tears can be distinguished based on their lesion pattern into radial, longitudinal, horizontal, circumferential, root lesions and bucket handle lesions [16]. Another way to classify meniscal tears, which is, prognostically, very valid, is in vascular (peripheral portion) and avascular (central portion) lesions (Figure 4).

The injuries of subjects under 40 are usually post-traumatic or associated with congenital meniscal conditions (e.g., discoid meniscus or meniscal cyst); on the contrary, meniscal lesions over 40 years of age are usually degenerative [17].

Furthermore, meniscal tears are closely associated with the development of knee osteoarthritis, and have a reciprocal relationship. A meniscal injury can, in fact, lead to the development of osteoarthritis, which, in turn, can itself cause meniscal lesions: undamaged menisci are difficult to find in a person affected by knee osteoarthritis. A torn meniscus stimulates the synovial tissue to produce pro-inflammatory cytokines that cause alterations to the meniscus matrix. The latter loses its biomechanical characteristics and undergoes morpho-structural changes up to its extrusion. Meniscal extrusion has traditionally been defined as meniscal tissue extending 3 mm or greater beyond the edge of the tibial plateau, excluding marginal osteophytes, measured on magnetic resonance imaging (MRI) [18]. Meniscal extrusion, in addition to painful symptoms, causes an increase in the mechanical stress of the articular cartilage, which facilitates the development of knee osteoarthritis. Furthermore, the cytokines released in the articular context cause damage both at the meniscal level and at the cartilage level, as the components of the extra-cellular matrix are similar. Therefore, a vicious circle is established, progressively leading to the development of osteoarthritis [19].

At the dawn of meniscal surgery, in the 1960s, the treatment for any type of meniscal injury was total meniscectomy, whether by open technique or by arthroscopy.

Later on, in the 1990s, clinicians discovered that total meniscectomy led to osteoarthritis within 5–10 years [20]. Several studies on cadavers from meniscectomies have observed increased contact pressures of up to 80–90%, which increases with progressively larger amounts of resection, and the augmented pressures experienced by articular cartilage can lead to premature cartilage wear [21,22]. In the knee, after a complete meniscectomy, the contact area is reduced by approximately 50%, resulting in a dramatic increase in load per unit area. Even a partial meniscectomy of just 15% has been shown to enhance the contact pressures to over 350% [23]. Thus, further studies were necessary to develop new meniscal sparing and repair techniques. Nowadays, meniscectomy is indicated only for small meniscal tears and with little or no regenerative potential [20]. In contrast, meniscal sutures and repairs have received a lot of attention in recent years: the aim of orthopedic surgeon, now, is to save as much meniscus as possible. The suturing techniques can be divided into inside-out, outside-in and all-inside [24,25,26]. The all-inside technique can be considered the gold standard for most meniscal tears, and the inside-out and outside-in techniques are good repair alternatives in selected meniscal tears, such as those of the anterior horn [27].

In order to increase the efficiency of these techniques, ancillary techniques have been proposed, such as needling, abrasion, trephination, gluing, synovial flaps and fibrin clots. In addition, an ever-increasing number of reconstructive techniques have been developed to restore the torn meniscus [28,29,30].

### 2.4. Clinical Evidence of Needing New Regenerative Strategies

The incidence of meniscal tears in young patients is constantly increasing. In fact, more and more adolescents and young adults are practicing contact sports or sports at risk of knee sprains and at a higher competitive level [31]. Thus, more patients undergo meniscal surgery at a young age, exposing the knee to many years of altered loads, inevitably leading to arthritis [32].

For this reason, there is a growing need for new reconstructive strategies that go beyond traditional meniscal surgery.

A possible constructive strategy could be a meniscal transplant with allograft. The meniscus, in fact, represents an ideal tissue for transplantation: it does not require vascular anastomosis and its cells are hidden by the immune system due to the avascular environment. Immunosuppressive therapy is therefore not required [33]. To date, meniscal transplantation is the only biological option for the symptomatic knee that is completely meniscectomized, stable and without signs of osteoarthritis (OA). It allows for good results for short-term symptoms, but it is not yet completely clear whether it protects in the long term from the development of osteoarthritis. Due to the length, cost and substantial rehabilitation of meniscal allograft transplantation, it is essential to select appropriate surgical candidates. It has been shown that appropriate candidates are young, healthy and active patients that are symptomatic after a prior meniscectomy and have failed further non-operative management [34]. A systematic review by Fanelli et al. found that patients greater than 35 years old had significantly worse patient reported outcomes when compared to their younger counterparts [35].

To date, four types of allografts have been used: cryopreserved, deep-frozen (fresh-frozen), fresh and lyophilized (freeze-dried) [36].

Fresh allografts, according to some authors, are rich in viable cells and increase the biomechanical integrity of the meniscus [37]. It is equally true that, from studies conducted on animals, the host cells replace the allogenic cells of the graft; hence, the use of a vital graft is still under debate [38].

A meta-analysis showed that this technique should no longer be considered experimental surgery, but a bridge to increase the time to arthroplasty [39]. Allograft fixation remains technically demanding, its retrieval is difficult and there is often a mismatch between the meniscus to be transplanted and the recipient knee. A retrospective review reported an overall failure rate of 29% for meniscal allograft transplantation (MAT) at 4–14 years post-op [40]. Meniscal transplantation in useful in the short-to-medium term for knee pain, whereas both a clear effect on cartilage preservation in the long-term and OA prevention has still not been demonstrated [41]. Another literature review shows that the allograft has an higher reoperation and failure rate than the available acellular meniscal scaffold [42].

Thus, the application of biomaterials for meniscal repair is desirable, as these scaffolds can mimic the meniscus and can support the biomechanical and cellular functions of this tissue.

## 3. Scaffolds

New biomaterial technologies lead to the development of meniscal scaffolds, which are then populated by the host cells recruited from the synovium and the remnants of the meniscal tissue; hence, a cell-free technique becomes a cell-based technique [43].

A biomaterial should possess, according to van Tienen et al., a series of very specific characteristics [44]:−It should promote cell migration;−It should be biomimetic, mimicking the architecture, tribology and mechanical characteristics of the native meniscus;−It should resist the forces that are applied and transmitted by the knee once the repopulating cells have produced an extracellular matrix;−It must be biocompatible, in order to avoid a reaction from a foreign body;−It must be slowly biodegraded, allowing for a progressive replacement from the host tissue;−It should possess a porosity that allows for the diffusion of nutrients and substances produced by cellular catabolism;−It must be easy to handle and easy to implant.

Several materials have been tested. Meniscal scaffolds can be divided into synthetic and biological (Table 1).

### 3.1. Synthetic Polymers

The synthetic scaffolds have the undoubted advantage of being manufactured with the desired shapes and sizes and with a precise porosity dimension and biomechanical characteristics. For example, Klompmaker et al. demonstrated that the optimal micropore size for colonization and cell growth should be 150–500 µm [45]. Another characteristic that can be modulated is the absorption rate, which varies with the composition of the scaffold [46]. To date, the most common polymers are polylactic acid (PGA) [47], poly-(l)-lactic acid (PLLA) [48], poly- (lactic-co-glycolic acid) (PLGA) [49], polyurethane (PU) [50], polyester carbon [51] and polycaprolactone (PCL) [52,53]. To facilitate the process of cell ingrowth, the polymer was provided with a porosity of ~80% and with a homogeneous distribution of larger interconnected pores (pore size, 150–355 μm), and the compression modulus at 20% compression was 300 kPa [54].

The possible complications that are more common in synthetic polymers than natural polymers are poor cell adhesion and the possibility of developing a foreign body reaction or aseptic inflammation, leading, therefore, to alter the joint architecture and, consequently, to worsen the functional outcomes [53].

Koller et al. tried to increase the biological activity of polymeric scaffolds by adding polyethylene terephthalate (PET) to PCL scaffolds: they showed that PET scaffolds increase the production of collagen II mRNA and the secretion of glycosaminoglycans [55] when added to a scaffold made of a hyaluronic acid/polycycaprolactone biomaterial. Through histological analysis, they demonstrated the colonization, proliferation, differentiation and attachment of meniscocytes. Baker and Mauck have produced scaffolds with aligned components with the use of electrospinning, improving the biomechanical properties and the ability of cell growth [56]. Fisher et al. modified the orientation of scaffold nanofiber composed of poly(ε-caprolactone) with fibers aligned in only one direction (0° or 90° with respect to the long axis of the scaffold) or circumferentially aligned (C) through electrospinning, and showed that the circumferential alignment of the cells seeded on the scaffold is increased and leads to a better mimicking of the cellular orientation of the native meniscus [57]. Koller et al. used a mix of PGA and PLGA (in a 75:25 ratio) to produce a meniscal scaffold implanted in an ovine model [55]. The histological examination showed a regenerated meniscal, like the native meniscus, but the scaffold did not prevent the development of arthritis. Kon et al. used PCL and hyaluronic acid scaffolds for complete and partial meniscal defects in goat knees [58]. Histological studies showed that the scaffolds were well integrated, well vascularized and invaded by host cells. Nevertheless, body weight bearing led to the extrusion of the scaffold and consequent arthritic degeneration [59]. A new promising scaffold is the Actifit implant (Orteq Sport Medicine, London, UK): an acellular scaffold composed of PU (20%) and PCL (80%). Studies on a limited number of patients have shown good integration, good patient satisfaction and an absence of cartilage lesions on imaging studies (Magnetic Resonance). The authors demonstrated how the functional scores were stable between months 12, 24 and 60. The ICRS cartilage score and mean meniscal extrusion were unchanged at the last follow-up (1.6 versus 1.6 and 2.41 versus 2.79). In all patients, the meniscal implant had an intermediate signal and reduced size on MRI. Despite an abnormal appearance on MRI suggesting that the meniscal scaffold is not fully mature after 5 years, the functional scores and cartilage status were stable after 5 years from implantation. However, the failure rate was still high, and the implant removal in patients with poor function did not improve the outcome [60,61,62].

### 3.2. Biological Scaffolds

The biological materials that have been used over time are the periosteal tissue [63], the perichondrium [64], the small intestine submucosa (SIS) [65], acellular porcine meniscal tissue [66,67] and bacterial cellulose [68]. Although these materials have a very high biocompatibility, some components are not suitable for tissue engineering, as their conformation and mechanical properties cannot be modified [25].

In order to increase the biocompatibility and biodegradability of the polymeric scaffolds, biopolymers, such as silk fibrous protein, can be used [69,70,71,72]. Fibrofix^TM^, a silk-derived meniscus scaffold, allows for cartilage protection in a sheep model of a partial meniscus replacement [73]. Gruchemberg et al. demonstrated how this scaffold was able to withstand the loads experienced during the implantation period. It did not cause an inflammatory reaction in the joint 6 months after surgery, and there were no significant differences in cartilage degeneration between the scaffold and the sham group. The compressive properties of the scaffold approached those of the meniscal tissue. However, the scaffolds were not always stably fixed in the defect, leading to cracking between the implant and host tissue or a total loss of the implant in three of the nine cases in each scaffold group. Thus, the fixation technique needs to be improved in order to achieve better integration into the host tissue [73].

Collagen or proteoglycans are excellent candidates for meniscal engineering, as they maintain a high biocompatibility and allow for the modification of the porosity texture and size the adaptation to the meniscus shape of the patient [26]. On the other hand, they have poor biomechanical characteristics and a more rapid degradation rate, compared to other ones, which could interfere with the complete replacement made by the host tissue [74].

Stone et al. demonstrated a substantial meniscal regeneration, following the use of copolymer scaffolds based on the collagen derived from a bovine Achilles tendon using 24 meniscectomized knees of skeletally mature dogs as an experimental model [75]. More recently, Rodkey et al. used a collagen scaffold in a controlled randomized trial in subjects with chronic and acute meniscal tears (the control group did not receive the meniscal scaffold). Biopsies performed one year after the implant demonstrated good scaffold integration in the chronic group and good functional outcomes, measured with validated outcome questionnaires, compared to the control group. Histological evaluations suggest that the implant supported the production and integration of the meniscus-like matrix as it was assimilated and reabsorbed. Subjects with acute lesions did not show an outcome difference [76]. Zaffagnini et al. conducted a cohort study with a minimum follow up of 10 years with good clinical results. In fact, pain, activity level and radiologic outcomes were significantly improved with the use of the medial collagen meniscus implant at a follow-up of at least 10 years compared with the partial medial meniscectomy alone [77]. Randomized controlled trials should be performed with a longer follow-up and with a larger patient population to confirm the effectiveness of collagen-based meniscal scaffolds.

Bacterial cellulose (BC) is a polysaccharide synthesized from Gluconacetobacter xylinus, and has numerous advantages in the production of scaffolds, as it possesses high biomechanical characteristics, a high hygroscopy and good biocompatibility [78]. Bodin et al. compared BC with collagen and a native pig meniscus. The compression modulus of BC was five times greater than the compression modulus of collagen, but still lower than that of the native pig meniscus [68].

### 3.3. Hydrogel Scaffolds

An interesting alternative is represented by the hydrogel scaffolds. Their semi-liquid nature allows for the generation of scaffolds with very precise geometries obtained from diagnostic images (i.e., MRI) [79]. Promising results have been reported with alginate and polyvinyl alcohol (PVA), which have excellent visco-elastic and biocompatible properties [80,81,82,83,84]. These materials are highly hydrophilic; thus, they possess good biomechanical properties.

Kobayashi et al. synthesized a PVA on rabbits, demonstrating cartilage tissue integrity 12, 18 and 36 months after the implantation of the scaffold. The author demonstrated that a PVA-H with a high water content exhibited viscoelastic behavior similar to the human meniscus. In addition, the friction coefficient of PVA-H against natural articular cartilage was also effective. In the animal experiment with rabbits, the lateral meniscus was replaced with an artificial meniscus in one side of the knee, and a lateral meniscectomy was performed in another side of each rabbit’s knee. The knees were evaluated after 1.5 years. In the results, the articular cartilage status of the knee joint with the implanted PVA-H meniscus was good, whereas the OA change progressed in the knee joint with the meniscectomy. In addition, neither wear nor a rupture of the PVA-H was observed [82]. Gogran et al. fabricated a 3D methacrylated gelatine (GelMA) meniscal scaffold using projection stereolithography. The authors then seeded the scaffolds with meniscal cells for two weeks in a chondrogenic environment and then implanted the scaffolds. They proved their non-toxicity and the correct induction of cell growth. Scaffolds in gelatin and chitosan have been created: their combined action increase the hydrophilic properties and the transfer of oxygen and nutrients [85].

Furthermore, hydrogel scaffolds were enriched with growth factors; in particular, platelet-rich plasma (PRP) and bone marrow aspirate concentrate (BMAC). They increased the neoangiogenesis, viability, proliferation and migration of fibrochondrocytes. Ishida et al. used a gelatin hydrogel (GH) as a PRP carrier in the knees of meniscectomized cones. They demonstrated how PRP stimulated deoxyribonucleic acid synthesis and ECM synthesis. Meniscal cells cultured with PRP showed an increased expression of biglicane and decorin mRNAs. Histological results showed that remnants of the gelatin hydrogels existed at 4 weeks, indicating that the hydrogels could control the release for approximately 4 weeks. The histologic scoring of defect sites at 12 weeks revealed a significantly better meniscal repair in animals that received PRP with GH compared with the other two groups [86,87].

### 3.4. Decellularized Meniscal Scaffolds

The decellularized meniscal tissue provides the correct microenvironment for the fibro-chondrocytes and preserves the correct meniscal anatomy [87]. However, the cell colonization of the scaffold is difficult; thus, the use of growth factors may be advisable. For example, a high concentration of bone morphogenetic protein-2 (BMP-2) stimulates the differentiation of MSC and can improve cell migration, as shown by Minehara et al., where a recombinant bone morphogenetic protein-2 (rhBMP-2) added to human menisci induced cell migration and increased proteoglycan production in vitro [88].

Sandmann et al. used sodium dodecyl sulfate (SDS) as the main product for the decellularization of human menisci and demonstrated that this method maintains the structures of collagen fibers. Mechanical tests demonstrated biomechanical characteristics that were very similar to the native meniscus, and histological analysis did not show the presence of native meniscal cells [89]. Stapleton et al. have studied a physical and chemical multiphase meniscal decellularization method, demonstrating a good structural conservation of proteins and biomechanics, without cytotoxic effects. They used a decellularized meniscal scaffold implanted in porcine knees. Histological, immunohistochemical and biochemical analyses of the decellularized tissue confirmed the retention of major structural proteins. There was, however, a 59.4% loss of glycosaminoglycans. The histoarchitecture was unchanged, and there was no evidence of expression of the major xenogeneic epitope, galactose-alpha-1,3-galactose. The biocompatibility of the acellular scaffold was determined using contact cytotoxicity assays. Decellularized tissue and tissue extracts were not cytotoxic to cells. Biomechanical properties were determined by indentation and tensile tests, which confirmed that biomechanical properties were maintained after decellularization [90]. Stabile et al. simultaneously used decellularization and oxidation to increase the porosity of the material. Some authors have applied sonication in order to obtain decellularized meniscal scaffolds. These scaffolds have good biomechanical properties, without immunogenic cells; on the other hand, sonication modifies the structure of the ECM proteins and the arrangement of the collagen fibers [91].

In order to improve the biomechanical properties, Yuan et al. developed a hybrid scaffold for regenerating the meniscus in a rabbit model, combining an acellular meniscus extracellular matrix (AMECM) and demineralized cancellous bone (DCB). The AMECM/DCB constructs demonstrated favorable mechanical properties and a promising capacity to promote fibrochondrocyte proliferation and GAG secretion [92].

### 3.5. Hybrid Meniscal Scaffold

In recent years, several researchers have developed meniscal scaffolds combining different biomaterials in order to optimize the mechanical and biological characteristics of each polymer [93]. For example, biological polymers, such as chitosan, collagen and gelatin, allow for excellent cellular interactions; on the contrary, synthetic polymers guarantee better biomechanical properties and a greater reliability in the degradation time [94,95]. Loading hybrid scaffolds with tissue-derived cells has the advantage of encapsulated cells replenishing the ECM loss and filling the defects as scaffold degradation occurs over time [96]. Furthermore, polymers combined with signal molecules can implement cell growth and colonization by creating a favorable microenvironment. For example, Li et al. built a hybrid scaffold using silk fibroin, polycaprolactone and a bioactive peptide: this scaffold protected cartilage and favored meniscal regeneration in an in vivo study using rats [97].

Thus, hybrid scaffolds are suitable for meniscal tissue engineering. The general design strategy consists of synthetic polymers as a supporting framework, with natural polymers more likely serving as an additive microenvironment to mimic extracellular microenvironments, whereas cells and bioactive factors may further assist in improving cell recruitment, proliferation and differentiation, and may ultimately improve regeneration [98].

**Table 1 pharmaceutics-13-01886-t001:** Relevant natural and synthetic polymer types used in meniscal tissue engineering.

Material	Advantages	Limitation	References
Synthetic			
Polilactic acid (PGA)	Excellent mechanical properties Bioresorbability	Potential adverse tissue reaction for polymer fragments	T. Murakami et al. [47];
Poly-(l)-lactic acid (PLLA)	High mechanical strength Thermal stability Tunable properties	Acidic products Autocatalytic degradation	A. P. Testa Pezzin et al. [48]
Poly-(lactic-co-glycolitic acid) (PLGA)	Tunable degradability Biocompatibility	Acidic byproducts	Y. Gu et al. [49]
Polyurethane (PU)	Good mechanical properties and cytocompatibility Thermoplasticity	Low reabsorption rate	T. De Coninck et al. [50]
Polyester carbon	Good mechanical properties	Cytotoxicity	J. Gopinathan et al. [51]
poly(ε-caprolactone)	Biocompatibility Biodegradability	Hydrophobicity Host cell low interaction	Z. Abpeikar et al. [52] R. T. C. Welsing et al. [53]
Polietilene + PCL	Increased secretion of glycosaminoglycans	Reduced cell adhesion	U. Koller et al. [55]
PU + PCL	Good integration, good clinical results	Possible implant extrusion	A.Leroy et al. [60] C.Baynat et al. [61] E. Bulgheroni et al. [62]
Biological			
Perichondrium	Good biocompatibility	Cartilage differentiation when in articular environment	C J Walsh et al. [63]
Small intestine submucosa (SIS)	Partial meniscal regeneration	Progression of joint damage	M P Bradley et al. [65]
Acellular porcine meniscal tissue	Excellent immunocompatibility	Absence of mechanical assessment	T. W. Stapleton et al. [67]
Bacterial cellulosa	Inexpensive, moldable and promotes cell migration	Poor mechanical properties	A.Bodin et al. [68]
Silk fibrous protein	Flexible processability Biocompatibility Capable of chemical modification Thermal stability Good mechanical strength	Immunogenicity Poor cell adhesion	A.Bandyopadhyay et al. [69] R. Yan et al. [70] S. E. C. Stein et al. [71]
Collagen	Cytocompatibility, capable of clinical use	Immunogenicity Weak mechanical strength	K. R. Stone et al. [75] W. G. Rodkey et al. [76]
Hydrogel			
Polyvvinil alcohol (PVA)	Excellent visco-elastic and biocompatible properties	Hard fixation method, tolerance of PVA-H	M. Kobayashi et al. [82] M. Kobayashi et al. [81]
Methacrylate genatine (GelMA)	Biocompatibility, biodegradability	Poor mechanical properties	S. P. Grogan et al. [85]

## 4. New Technology and Future Perspective

Three-dimensional (3D) printing is a very interesting method for meniscus repair because it allows for a patient-specific customization of the scaffolds [96]. A promising 3D printed fibrous polycaprolactone (PCL) scaffold has been created in order to direct cell differentiation or to induce cell homeostasis. This study employed MSCs seeded on this PCL scaffold for in vitro studies to demonstrate their differentiation into fibrochondrocytes, whereas the scaffold was used without cells in the large animal study. A partial meniscus replacement in sheep using 3D printed PCL scaffolds demonstrated the deposition of zone-specific type I and II collagen. This study highlighted the importance of growth factor delivery systems in an attempt to mimic the zonal variation in the meniscus [99]. Other authors have implemented their scaffolds based on this study [100].

In general, bioprinting technologies mainly applied in meniscus or cartilage tissue engineering can be classified as follows: 3D plotting/direct ink writing, stereolithography (SLA), selective laser sintering (SLS), fused deposition modeling (FDM) and extrusion-based bioprinting [101]. Tissue menisci scaffolds produced by extrusion bioprinting techniques have high yields and excellent structural integrity, and this technique is most applied in meniscal regeneration [98]. Numerous polymers have been extensively studied to serve as bio-inks in 3D printing for tissue engineering. Bio-inks, mainly hydrogels that contain cells and various biological components, play an important role in creating a compatible microenvironment for cellular activity [102]. Recently, tissue-specific meniscal dECM (me-dECM) bio-inks based on 3D cell printing were designed by Chae et al. (2021) and helped to preserve the complexity of the natural ECM, thus demonstrating the potential for tissue regeneration and special biological functions. This printable bio-ink supported the proliferation and differentiation of encapsulated stem cells in vitro [103]. The inkjet technique, which is suitable for printing cellularized scaffolds, can be used to print customized cellularized menisci in order to embed MSC in a printed meniscus [104].

Studies have indicated that the transduction of biomechanical stimulation to molecular signals can regulate cell differentiation and maturation, highlighting the important role of mechanical stimulation in meniscal development, growth and health [105,106].

The three most common biomechanical stimuli are hydrostatic pressure, direct compressive loading and tension stimulation.

Studies have shown an increased extracellular matrix formation in chondrocytes under hydrostatic pressure conditions. For example, a 1.3-fold increase in GAG was found in chondrocytes exposed to hydrostatic pressure compared to static controls [107]. Many studies have demonstrated the importance of hydrostatic pressure for the chondrogenic differentiation of mesenchymal stem cells (MSCs). The application of continuous or cyclic hydrostatic pressure can increase the extracellular matrix deposition, type II collagen production and chondrogenic differentiation of MSCs [108,109,110].

Some studies show that matrix synthesis is increased by dynamic compression stimulation; for example, with a compressive stimulation of 0.5 kPa, the collagen content of chondrocytes was increased 1.5-fold compared with free-vessel controls [111].

In the study by Lee et al., uniaxial tensile loading increased the matrix formation in explanted chondrocytes by 33% [112].

A new component used for the construction of meniscal scaffolds using a 3D printer is silicone. Silicone implants have conventionally been fabricated using indirect methods of silicone casting and molding that are expensive and time consuming. Luis et al., using an extrusion method, have developed a scaffold that demonstrates mechanical properties like the conventionally hot molded meniscus. In theory, the meniscal scaffold should provide appropriate biomechanical functions after implantation to shield the cells from damaging compressive or tensile forces, to maintain their integrity of shape (no shrinkage, etc.), mechanical stability and force over the defect area until enough host tissue has regenerated and to produce mechanical stimuli to promote tissue regeneration [113].

In a subsequent study, they demonstrated that the silicone 3D printing process does not permanently change the physical, biochemical or mechanical properties of the 3D printed silicone meniscus [114].

In conclusion, the excellent ability of 3D bioprinting technology to precisely control the fiber diameter, orientation and microarchitecture endows the resulting constructs with promising mechanical properties and favorable biological functions. Therefore, it is one of the most attractive and promising technologies for tissue engineering applications, particularly meniscal regeneration [98].

In recent years, cellular self-assembly began to gain recognition and support in the generation of articular cartilage [115]. Lately, the concept of self-assembly is also being used for meniscal engineering, with some advantages. First, the natural synthesis and adherence to cartilage ECM generate the most bioactive microenvironment of any approach. Second, the all-biologic construct greatly increases the chances of integration with the host tissue. Third, the lack of a scaffold diminishes further contributions to an immune response. However, the use of allogeneic cells in any vascularized setting presents a risk of acute immune rejection. Autologous cell sources would obviate this risk while having their own donor challenges, such as two-stage surgery and donor site morbidity. In addition, the removal of any degradation products minimizes potential toxicity and allows for a greater cell viability. Moreover, stress-shielding effects exerted by scaffolds are mitigated. More robust and homogeneous mechanical stimulation is possible during tissue development, which is particularly pertinent for engineering the knee meniscus. Lastly, since self-assembled tissue has a continuous ECM, it may possess a better modeling capacity in response to catabolic exogenous agents [116,117].

## 5. Conclusions

The meniscus is a fundamental structure for knee homeostasis. Its damage can alter the biomechanics of the joint and can lead to the development of OA within a few years. Therefore, new technologies are crucial to develop novel substitutes that faithfully resemble the biology and biomechanics of the native meniscus. The optimization of the treatment of meniscal injuries remains a challenge of paramount importance, which requires advanced therapeutic strategies based on the recognition of the pathophysiology of the damage and the restoration of joint biomechanics in order to avoid the arthritic evolution. There is, in fact, the need for therapies that are applicable on a large scale, which can restore both the biomechanical characteristics and the capabilities in terms of molecular signaling.

The optimal scaffold should be characterized by many biophysical and biochemical properties, as well as bioactivity, in order to ensure an ECM-like microenvironment for cell survival and the differentiation and restoration of the anatomical and mechanical properties of the native meniscus. In general, polymer selection remains a knotty problem, as natural polymers exhibit a better biocompatibility and inherent bioactivity, whereas synthetic polymers have superior mechanical properties. However, single-polymer scaffolds are insufficient in mimicking the composition and structure of the meniscus [98]. In addition, each zone of the meniscus has a different architecture. For this reason, the use of the 3D printer would seem to be particularly promising, as it is able to differentiate the structure of the meniscal substitute itself, and fiber-reinforced constructs are also advantageous for driving ECM deposition and cell distribution. In addition, 3D printed meniscal scaffolds can be seeded with different cells and signal molecules to optimize joint cross-talk in vivo.

Hence, 3D printing seems to be the most promising strategy, currently, in meniscal scaffold creation.

To date, we are not able to recreate a meniscus substitute that has the ideal characteristics. Certainly, in order to find a biomaterial that meets all of the characteristics sought, we will have to investigate, in addition to biomechanical knowledge, the interactions with the joint microenvironment, which is an essential element in creating a biomaterial that integrates with the surrounding structures. Endogenous stem cells respond to biochemical signals, migrate to damaged sites, differentiate into somatic cells and restore their morphology and function [118]. Meniscal scaffolds would appear to hinder cell migration because of the dense extracellular matrix [119]. Therefore, polymeric materials that are able to induce cell migration and provide a suitable microenvironment for cell adhesion and proliferation have been shown to be promising candidates for use in meniscal regeneration. For example, Zhang et al. demonstrated that a 3D printed PCL scaffold with an average pore size of 215 μm significantly enhanced the colonization of seeded cells and further improved meniscal regeneration [120]. Apart from modifying the porosity of the scaffold, an incorporation of chemotactic chemokines could be a promising avenue [121,122,123].

The development of new biomaterials remains the basis for tissue engineering of the meniscus. What is currently missing is the clinical application of these biomaterials on a large scale and their validation in randomized controlled trials.

The new technological advances in recent years, such as 3D bioprinting and mesenchymal stem cells management, will probably lead to an acceleration in the design, development and validation of new and effective meniscal substitutes.

## Figures and Tables

**Figure 1 pharmaceutics-13-01886-f001:**
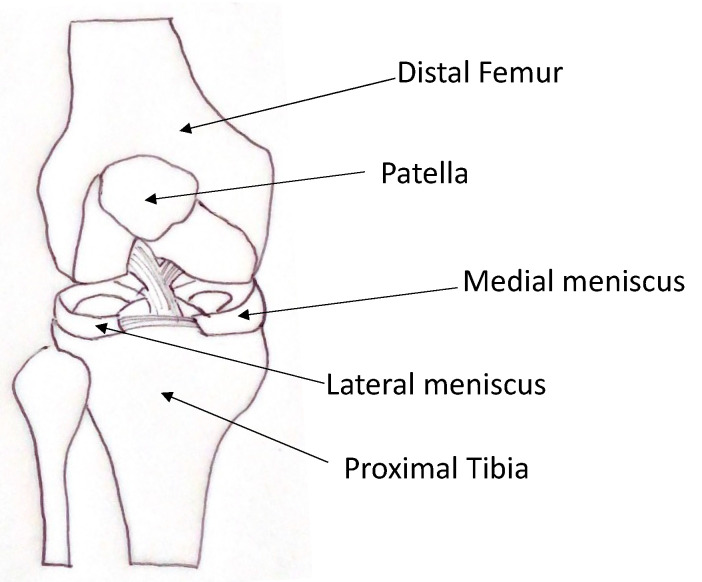
Schematic representation of knee anatomy with menisci.

**Figure 2 pharmaceutics-13-01886-f002:**
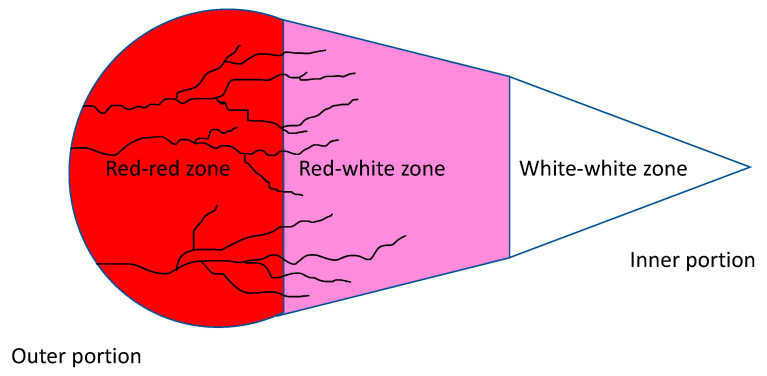
Schematic cross section diagram meniscus representing vascular features of meniscus vascular zones. Blood vessels are represented in black.

**Figure 3 pharmaceutics-13-01886-f003:**
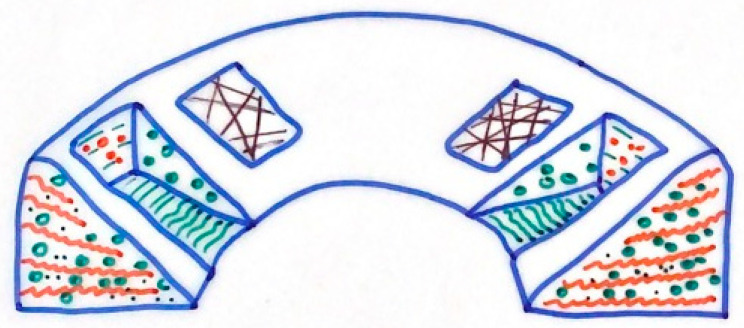
Schematic diagram of ultrastructure of collage fiber of meniscus. Circumferential fiber is represented in green and radial fiber is represented in red. Irregular superficial fibers are represented in black.

**Figure 4 pharmaceutics-13-01886-f004:**
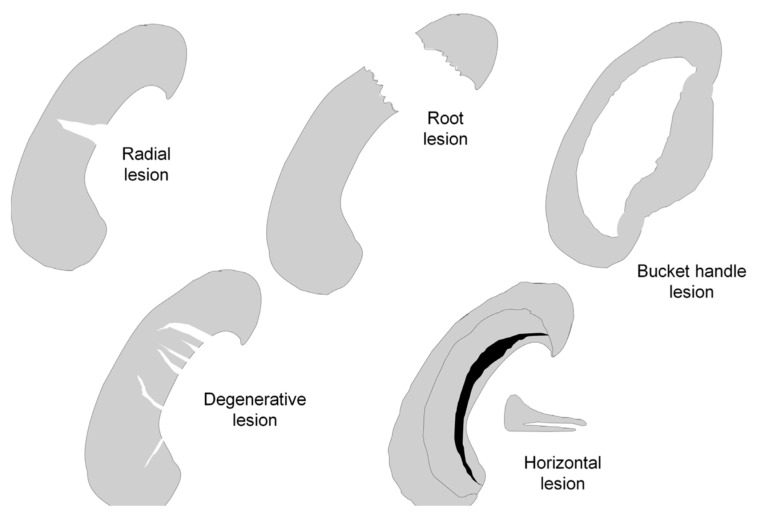
Schematic diagram of different type of meniscal lesions. In this illustration, the meniscus is viewed from above, with the lower portion of the figure corresponding to the anterior horn of the meniscus.

## Data Availability

All information is contained in scientific publications downloadable from Pubmed.

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
