# Peer review of "Biomaterials and Meniscal Lesions: Current Concepts and Future Perspective"

_pharmaceutics, 2021, doi:10.3390/pharmaceutics13111886_

Round 1

Reviewer 1 Report

The work presented in this manuscript is interesting and well-suited for publication in pharmaceutics. The authors review literature on the use of different biomaterials used in treatment of lesions in knee meniscus. However, the work done is still lacking and there are a few major points which must be addressed before this publication can be considered complete.

  1. The introduction is too short and more elaboration should be given on the importance and relevance of the topic being discussed in this review. Also, a short overview of the sections (e.g. types of scaffolds) being discussed in the review should be included in the introduction, this is completely missing.
  2. A novel biomaterial used for knee meniscus implants is silicone, this is a hot emerging area of research in this field and should be included either as a section or at least discussed in the review. Here are some references you can consider citing (https://doi.org/10.3390/polym12092136 and https://doi.org/10.3390/polym12051031).
  3. The entire review is severely lacking in appropriate figures to illustrate the literature that is being discussed. There should be figures illustrating important sections being discussed such as the types of scaffolds and new technologies such as 3D printing being employed in this field.
  4. The title for section 3.2 has a spelling error. It should be “biological scaffolds”.
  5. The discussion on the different scaffolds in also lacking in the comparison of the advantages and disadvantages of different biomaterials used as scaffolds. A deeper analysis and discussion of which type of biomaterial will be more suitable for certain application will also be interesting to include in the review.
  6. The review is lacking a table summarizing all the different types of scaffolds, biomaterials being used, fabrication techniques, and specific references where you can find the literature. This will provide for readers a good summary of the entire review and improve comprehension and readability of the review.

Author Response

Dear reviewer, thank you so much for your suggestions.

  • We changed introduction as you suggested, by adding information about topics taken into consideration in the manuscript.
  • We have added information regarding silicone-based meniscal replacements, as requested, using the suggested references.
  • We added figures and diagram in order to simplify the reader's understanding of the manuscript. Regarding 3D printing illustrations, thank you for the suggestion, but due to the limited time we are not able to request the copyright.
  • We changed to “biological scaffold” as requested
  • We changed the discussion part, delving into required topics, making dissertation about different biomaterials.
  • We have added a summary table with the characteristics of biomaterials.

Reviewer 2 Report

Lombardo et al present a review of meniscus injury and repair using biomaterial scaffolds, with an emphasis on more recent techniques and trials. The authors provide a very clear and well written introduction to meniscus structure/function, the types of injuries it is susceptible to, and the resultant clinical challenges. They then discuss implantable material techniques, logically organized into familiar categories (grafts, synthetic scaffolds, biologic scaffolds, etc). There is a broad overview of materials, which is useful to introduce others to the field. However, the main concern throughout the Scaffolds sections is the lack of assessment provided for the various scaffold materials. Outcomes are referred to as “good” or “bad,” with very little context of what that means. For example, what constitutes favorable integration or mechanical properties, and compared to what? This review should include additional interpretation of study results.   

  1. Minor edit: consider using fewer paragraph breaks. Many paragraphs are only 1-2 lines which is a little distracting. (e.g. the beginning of section 3.)
  2. A figure showing the meniscus in its anatomical location and defining the zones of vascularization, ECM organization, and/or innervation would be very beneficial for the reader. A similar approach would be ideal for the current Figure 1, as it’s not clear how the meniscus is oriented in these various injuries for a scientific reader outside of the field.
  3. Please define meniscal extrusion when it is mentioned in the text.
  4. Not necessary, but just a suggestion: year ranges would complement or replace the history being provided. (i.e. when was the dawn of meniscal surgery and “later on?”)
  5. Additional commentary on the limitations of grafts would be helpful. Donor source & availability, immune response, etc.
  6. Are the complications in 3.1 specific to the synthetic polymers listed? Does the FBR occur to meniscus scaffolds made from these materials? It’s unclear how the limitations the authors mention manifest in meniscus repair.
  7. Please also provide additional description of the physical properties of the synthetic materials being focused on – what is their shape, microstructure, mechanical properties, etc? Which properties are most important to functional or histologic remodeling?
  8. Quantitative metrics vs control groups should be mentioned when discussing results of other studies. Results are described as “good” though it’s very difficult to interpret what that means.
  9. Why are silk protein scaffolds listed among synthetic scaffolds? Wouldn’t this be biologically derived?
  10. Reference 49 on line 204 is apparently an in vitro model though the text suggests it’s in vivo. Please double check the citations and provide more context. It’s often difficult to tell what’s relevant to in vivo studies.
  11. What models are used to evaluate these materials in vitro and in vivo? i.e. what in vivo injury models are used to study and what metrics are used to show efficacy?
  12. Is there any evidence that the synthetic scaffolds listed are degraded and remodeled into tissue?
  13. Representative histology or other images from the cited studies would be hugely beneficial. This is important to orient the reader with examples of integration and degradation processes. For collagen, the authors state that it may not be suitable due to rapid degradation, yet later cite studies in which there are 1 year biopsies showing favorable integration. It’s difficult to visualize this; over 1 year is not generally considered a rapid degradation timeline.
  14. I suggest rephrasing the notion that self-assembled cellular constructs are scaffold free. Cells are cultured for months in the cited study and are shown by those authors to generate substantial ECM in the process. That ECM, though generated in vitro, would fall into the biologics class. And finally, cell source is a critical consideration the authors should mention here. Allogeneic cells in any vascularized environment present an acute immune rejection risk. Autologous cell sources obviate this risk, though having their own donor challenges.

Author Response

Dear reviewer, thank you so much for your suggestions.

We have reworded the section on scaffolds by adding critical analyses of the results of the studies shown, as you suggested

  1. We changed the text using few paragraph brakes, in order to facilitate the reader's understanding by avoiding distractions
  2. We added figures and diagram in order to simplify the reader's understanding of the manuscript
  3. we have deepened the caption of figure 1, making it more usable for the reader
  4. We defined what meniscal extrusion is and we have added the appropriate bibliographic reference
  5. We added dates and timelines
  6. We added data about meniscal allograft transplantation and its limitation.
  7. We added data about meniscal scaffold complication.
  8. We added data about physical properties of the synthetic materials as suggested.
  9. We added the results of the analyzed studies by quantifying and describing them.
  10. We have moved the discussion of silk-derived meniscal substitutes to the biopolymers section.
  11. Reference 49 is about in vivo model
  12. It depends on the study considered and the model used. If animal model: histological study. If human model, the histological study is added to the clinical evaluation. We have added in the text the data related to the results and methodologies of the studies. Histological studies showed that the scaffolds were well integrated, well vascularized and invaded by host cells. (ref 59). Another study showed colonization, proliferation, differentiation, and attachment of meniscocytes of meniscocytes (ref 55) Little evidence about remodeling.
  13. In the case of the study cited as reference number 76, since it was on a human model, a single control sampling per patient was performed after a reasonable period to expect tissue degradation and integration. We changed sentence about collagen degradation time. Regarding the illustrations, thank you for the suggestion, but due to the limited time we are not able to request the copyright.
  14. We made changes for the self-assembling meniscal construct as you suggested.

Reviewer 3 Report

This review aims to summarize current and possible future therapeutic and surgical options for meniscal lesions. This is certainly an important topic, as the menisci are of high importance for the biomechanics of the knee and menisectomy is most often associated with the development of osteoarthritis of the knee. I highly appreciate your effort to summarize the current state of clinical and experimental concepts as well as limitations of several treatments on this most important matter.

The review is, in my opinion, conclusive, well performed and well written. I especially like that the biomechanics of the menisci is described first, to make it easy to the reader to understand the need of menisci-preserving treatments. I only have some small points I would like to address and suggest that the paper is accepted with minor revision.

Major comments:

Compared to the quiet informative review, the abstract is missing information of the key findings of the review. I suggest that you rewrite the abstract and add specific details about the presented biomaterials as well as the conclusion you draw from your findings. I also suggest that you add more specific keywords.

You sometimes switch between present and past tense when presenting findings of other studies (“A meta-analysis shows” vs. “Klopmkaer et al demonstrated”. I suggest that you stick to one style.

It seems to me that you sometimes use an Oxford comma, but not always. Please use one style, chosen according to the journals policy, throughout the hole manuscript.

Minor comments:

L27: Important point, could you add some information on the importance of incongruence of joints to allow transportation of the synovial fluid? The menisci aid this fluid transport and, at the same time, increase congruency to decrease contact pressure. I think it is important to comment on the higher contact pressure after the removal of the menisci, as this is a key factor in the development of osteoarthritis.

L31: “Given the [difficult/poor] vascularization”

L51: “joinT”

L77: Can you explain why this fiber arrangement is biomechanically of advantage? L86 sums it up quiet well, but I think you could add more details.

L112: “leading to”

L230: Could you please specify “good clinical results”?

L259: Could you add some information on the importance of physiological biomechanics stimulation (like wall shear stress) on the differentiation of MSCs seeded on scaffolds to this paragraph? Or perhaps extend the information given in L340 and add studies investigating favorable mechanical conditions for tissue development?

L261: “However, the cell”

L345: I would like to see the conclusion more precise and supported by evidence. What is needed for the transition of new biomaterial into clinical application, besides RCTs? Which problems need to be solved in future research? How do you know that new advances will lead to effective meniscal substitutes?

Author Response

Dear reviewer, thank you so much for your suggestions.

  • We changed the abstract as suggested, adding insights on biomaterials treated in the manuscript and the conclusions draw from findings.
  • We changed the keywords as suggested
  • We have unified verbal tenses as suggested
  • We have eliminated the Oxford comma
  • L27 We added a statement regarding higher contact pressure after the removal of the menisci at L 126. And a statement regarding incongruence and lubrication at line 77
  • L31: we added “poor” to vascularization
  • L51: we fixed join”T”
  • L77: we added a couple of statements in order to deep the explanation of fibers arrangement
  • L112: we fixed leading to
  • L230: we added a statement to explain what “good clinical result means”
  • L259 and L340: we added, as you suggested, information about studies investigating favorable mechanical conditions for tissue development
  • L261: we fixed the uppercase “T”
  • L345: we changed conclusions ad you suggested

Reviewer 4 Report

Pharmaceutics (Ms. No.; pharmaceutics-1391191)

              This review summarizes the biomaterials focusing on meniscal lesions. Beneficial therapy for meniscal lesions is an important issue around the world so this review should be interested in the readers of Pharmaceutics. Just for reference, I make comments as below.

1) Please carefully confirm the English spelling and the typing error.
  ex) Page 2, Line 47: I found “are are”.

2) To help the readers understand, I recommend inserting the anatomical illustration of the knee joint as Figure 1.

3) If you are possible, please make a Table summarizing up the key points about scaffolds as a Table.

Author Response

Dear reviewer, thank you so much for your suggestions.

  • We erased one “are”
  • We fixed typos mistakes
  • We have added a summary table with the characteristics of biomaterials
  • We added figures and diagram in order to simplify the reader's understanding of the manuscript.

Round 2

Reviewer 1 Report

The comments have been sufficiently addressed and is now ready for publication.

Reviewer 2 Report

The authors have very thoroughly addressed the reviewer comments. All that remains are some typos and grammatical errors, especially in the newly added text . (e.g. the "I" is missing from immunogenicity in the table)